# Morphological and Phylogenetic Analyses Reveal Four New Species of *Acrodictys* (*Acrodictyaceae*) in China

**DOI:** 10.3390/jof8080853

**Published:** 2022-08-15

**Authors:** Shi Wang, Rongyu Liu, Shubin Liu, Zhaoxue Zhang, Jiwen Xia, Duhua Li, Xiaoyong Liu, Xiuguo Zhang

**Affiliations:** 1College of Life Sciences, Shandong Normal University, Jinan 250358, China; 2Shandong Provincial Key Laboratory for Biology of Vegetable Diseases and Insect Pests, College of Plant Protection, Shandong Agricultural University, Taian 271018, China

**Keywords:** morphology, new taxa, phylogeny, *Sordariomycetes*, taxonomy

## Abstract

During our ongoing survey of dematiaceous hyphomycetes associated with dead branches in tropical forests, eight *Acrodictys* isolates were collected from Hainan, China. Morphology from the cultures and phylogeny based on partial small subunit (SSU), entire internal transcribed spacer regions with intervening 5.8S (ITS), partial large subunit (LSU) of rRNA gene, partial beta-tubulin (tub2), and partial RNA polymerase II second largest subunit (rpb2) genes were employed to identify these isolates. As a result, four new species, namely *Acrodictys bawanglingensis* sp. nov., *A. diaoluoshanensis* sp. nov., *A. ellisii* sp. nov., and *A. pigmentosa* sp. nov., are introduced. Illustrations and descriptions of the four taxa are provided, along with comparisons with closely related taxa in the genus. For facilitating relative studies, an updated key to all accepted species of this genus is also compiled.

## 1. Introduction

*Acrodictys* M.B. Ellis was erected by Ellis and typified by A. *bambusicola* M.B. Ellis [1]. In the protologue, this genus was characterized by globose to subglobose, uniformly pigmented, angularly, or obliquely septate conidia. This broad, generic circumscription of *Acrodictys* has been followed for nearly 40 years until recent segregations starting at the beginning of this century [2,3,4].

*Acrodictyella* W.A. Baker and Partr., typified by *A. obovata* W.A. Baker and Partr., was considered the first genus to accommodate those species similar to *Acrodictys* but characterized by producing hyaline, muriform conidia, which secede well before maturation and become pigmented sometime after their release [2]. Based on conidial morphology, conidiogenesis, and conidial secession, Baker et al. [3,5] and Baker and Morgan-Jones [4] divided the genus *Acrodictys* sensu lato into four genera, viz. *Acrodictys* sensu stricto, *Junewangia* W.A. Baker and Morgan-Jones, *Pseudoacrodictys* W.A. Baker and Morgan-Jones and *Rhexoacrodictys* W.A. Baker and Morgan-Jones. The *Acrodictys* sensu stricto is mainly characterized by macronematous, mononematous, cylindrical, unbranched or infrequently branched, percurrently proliferating conidiophores, and muriform conidia [6,7]. Based on morphological and phylogenetic analyses and typified by the genus *Acrodictys* sensu stricto, *Acrodictyaceae* J.W. Xia and X.G. Zhang was introduced [7].

Species in *Acrodictys* sensu stricto are saprobic on dead branches. They are thought to be wood-decaying fungi, promoting the carbon cycle in the ecosystem by converting cellulose, hemicellulose, and lignin into inorganic substances [7]. Although having a worldwide distribution, they are mainly recorded from tropical areas, such as Brazil, Thailand, and Mexico (https://www.gbif.org, accessed on 25 June 2022). Hainan Province (18°10′–20°10′ N,108°37′–111°05′ E) is an island in southern China. Its annual mean temperature is 22–27 °C, and its annual precipitation is 1000–2600 mm. It is a typical tropical rainforest climate, suitable for the growth and reproduction of wood-decaying fungi.

In China, 12 isolates representing 9 species (*A. bambusicola*, *A. elaeidicola*, *A. fluminicola*, *A. globulosa*, *A. hainanensis*, *A. liputii*, *A. malabarica*, *A. peruamazonensis,* and *A. porosiseptata*) have been documented from tropical areas in Hainan and Yunnan Provinces [6,7]. The samples in this study were collected from the Bawangling National Nature Reserve and Diaoluoshan National Nature Reserve in Hainan Province. Dead branches of unidentified trees were collected in tropical rain forests dominated by *Lauraceae* and *Fagaceae* trees, and the fungi in pure culture were isolated from these dead branches. Fungi from these axenic cultures were identified based on classic morphological and modern molecular approaches. As a result, four new species are described and illustrated herein in the genus *Acrodistys* sensu stricto.

## 2. Materials and Methods

### 2.1. Isolation and Morphological Observation

Samples were collected from the Bawangling National Nature Reserve and Diaoluoshan National Nature Reserve, Hainan Province, China (108°37′–117°50′ E, 3°58′–20°20′ N), and taken to the laboratory in plastic bags. The samples were placed in Petri dishes with moist filter paper and cultured in an incubator at 25 °C for 1–2 weeks. The samples were examined using a stereomicroscope (SZX10 model, Olympus, Tokyo, Japan, https://www.olympus-lifescience.com.cn/zh/microscopes/, accessed on 25 June 2022) to locate sporulating structures. Isolations were made with single spore methods according to the following steps: Spores were suspended with sterilized water and spread onto PDA (PDA: 200 g potato, 20 g dextrose, 20 g agar, 1000 mL distilled water, pH 7.0) plate and incubated for one day under a biochemical incubator. After germination, colonies were transferred to a new PDA plate to obtain a pure culture. After 30 days of incubation, morphological characters were recorded. Photographs of the colonies were taken on the 30th day using a digital camera (G7X model, Canon, Tokyo, Japan, https://www.canon.com.cn/, accessed on 25 June 2022). The micromorphological characteristics of colonies were observed using two stereomicroscopes (SZX10 model and BX53 model, Olympus, Tokyo, Japan, https://www.olympus-lifescience.com.cn/zh/microscopes/, accessed on 25 June 2022), both fitted with a digital camera (DP80 model, Olympus, Tokyo, Japan, https://www.olympus-lifescience.com.cn/zh/microscopes/, accessed on 25 June 2022). Sporulating structures were mounted in water for microscopic observation and photomicrography. At least 50 mature conidia and conidiophores were measured using the cellSens software (Olympus, Tokyo, Japan), and images were used for further processing with Photoshop ver. CS5 (Adobe Systems, San Jose, CA, USA, https://www.adobe.com/cn/, accessed on 25 June 2022). The cultures were stored in 10% sterilized glycerin and sterile water at 4 °C for further studies. Specimens were deposited in the Herbarium Mycologicum Academiae Sinicae, Institute of Microbiology, Chinese Academy of Sciences, Beijing, China (HMAS), and the Herbarium of the Department of Plant Pathology, Shandong Agricultural University (HSAUP). Living cultures were deposited in the Shandong Agricultural University Culture Collection (SAUCC). New taxa were submitted to the taxonomic database MycoBank (http://www.mycobank.org, accessed on 25 June 2022) and obtained relative deposition numbers.

### 2.2. DNA Extraction and Amplification

Genomic DNA was extracted from fungal mycelia grown on PDA, using a modified cetyltrimethylammonium bromide (CTAB) protocol as described in Guo et al. [8]. Five loci, namely the internal transcribed spacer regions with intervening 5.8S rRNA gene (ITS), partial large subunit (LSU) rRNA gene, the partial small subunit (SSU) rRNA gene, the partial RNA polymerase II second largest subunit (rpb2), and part of the beta-tubulin gene region (tub2) were amplified with the primer pairs and polymerase chain reaction (PCR) processes listed in Table 1.

PCR was performed using a thermocycler (Mastercycler X50, Eppendorf, Hamburg, Germany, https://www.eppendorf.com/CN-zh/, accessed on 25 June 2022). Amplifications were performed in a 25 μL reaction volume, which contained 12.5 μL 2 × Taq Plus Master Mix II (Vazyme, Nanjing, China), 1 μL of each forward and reverse primer (10 μM) (Tsingke Company Limited, Qingdao, China), 1 μL template genomic DNA (approximately 1 ng/μL), and 10 μL distilled deionized water. PCR products were visualized through 1% agarose gel electrophoresis. Sequencing was bidirectionally conducted by a company (Tsingke Company Limited, Qingdao, China), and a consensus was obtained using MEGA 7.0 [9]. All sequences generated in this study were deposited in GenBank under the accession numbers listed in Table 2.

### 2.3. Phylogenetic Analyses

Novel sequences were generated from the eight strains in this study, and all available reference sequences of *Acrodictys* species were downloaded from GenBank [7,10]. No rpb2 sequences are available in GenBank, and only two sequences are available for tub2. Multiple sequences for ITS, LSU, and SSU were aligned and concatenated using an online program (MAFFT v.7.11, http://mafft.cbrc.jp/alignment/server/, accessed on 25 June 2022) [11], with default settings and manually corrected where necessary (Appendix A). To establish the identity of the isolates at a species level, phylogenetic analyses were conducted for each locus and for a combination of all three loci (ITS, LSU, and SSU), based on maximum likelihood (ML) and Bayesian inference (BI). The best evolutionary model for each partition was determined using MrModeltest v. 2.3 [12] and then incorporated into the BI analyses. ML and BI were run through the CIPRES Science Gateway portal (https://www.phylo.org, accessed on 25 June 2022) using RaxML-HPC2 on XSEDE 8.2.12 (Heidelberg, Germany) [13,14] and MrBayes on XSEDE 3.2.7a (Stockholm, Sweden), respectively [15,16,17]. ML was performed using the default parameters, and BI was carried out using the rapid bootstrapping algorithm with the automatic halt option. Bayesian analyses included four parallel runs of 5,000,000 generations and a sampling frequency of 500 generations. The burn-in fraction was set to 0.25, and posterior probabilities (PP) were determined from the remaining trees. The resulting trees were plotted using FigTree v. 1.4.4 (http://tree.bio.ed.ac.uk/software/figtree, accessed on 25 June 2022) and edited with Adobe Illustrator CS6.0 (https://www.adobe.com/cn/, accessed on 25 June 2022).

## 3. Results

### 3.1. Phylogenetic Analyses

Eight strains of *Acrodictys* were isolated from the dead branches of unidentified trees in Hainan Province, China. The alignment of ITS, LSU, and SSU sequences was composed of 21 strains (including the 8 new strains) of *Acrodictys* and *Fluminicola saprophytica* (MFLUCC 15-0976) as the outgroup taxon, and 2419 characters, viz. 1–592 (ITS), 593–1419 (LSU), and 1420–2419 (SSU). Of these characters, 2089 were constant, 94 were variable and parsimony uninformative, and 236 were parsimony informative.

The Bayesian analysis lasted for 65,000 generations, resulting in 1301 total trees, of which 976 trees were used to calculate the posterior probabilities. The ML tree topology is consistent with that from the BI analyses, and therefore, the ML tree is presented with BI posterior probabilities being plotted together (Figure 1).

The 22 strains are assigned to 15 species clades based on the three-locus phylogeny (Figure 1). The involved eight strains represent four new species, namely *Acrodictys bawanglingensis* (Figure 2), *A*. *diaoluoshanensis* (Figure 3), *A. ellisii* (Figure 4), and *A*. *pigmentosa* (Figure 5), forming a clade together with *A. bambusicola* (MLBS = 96% and BYPP = 1.0).

### 3.2. Taxonomy

#### 3.2.1. *Acrodictys bawanglingensis* S. Wang, J.W. Xia, X.Y. Liu, and X.G. Zhang, sp. nov.

MycoBank No. 844582

Etymology―The epithet *bawanglingensis* is named after the Bawangling National Nature Reserve where the holotype was collected.

Type―China, Hainan Province: the Bawangling National Nature Reserve (109°03′–109°17′ E, 18°57′–19°11′ N), on the dead branches of unidentified trees collected in tropical rain forests dominated by *Lauraceae* and *Fagaceae* trees, 19 May 2021, R.Y. Liu, holotype HMAS 352233, ex-holotype living culture SAUCC 1342.

Description―Asexual morph on PDA: Mycelia are white to pale brown, floccose cottony, reverse black. Conidiophores are macronematous, mononematous, erect, unbranched, straight or flexuous, thick-walled, smooth, dark brown at the base, paler toward the apex, septate, 60.0–120.0 × 4.5–6.5 μm. Conidiogenous cells are monoblastic, integrated, terminal, determinate, cylindrical, pale brown to brown, and smooth. Conidia are solitary, muriform, obovoid to obpyriform, pale brown to brown, 18.0–26.0 × 10.0–16.0 μm, usually with 3 transverse septa and 1–3 longitudinal septa, slightly constricted at the septa, with conspicuous pores in the septa, and truncate at the base. Chlamydospores were not observed. Sexual morphs are unknown.

Culture characteristics―Colonies on PDA are flat with an entire margin, attaining 25.0–30.0 mm in diameter after 14 days at 25 °C, with a growth rate of 1.5–2.5 mm/day, a greenish-brown color, and with a layer of white aerial hyphae on the surface. Colonies on MEA are flat with an entire margin, have a generally mouse-gray color, and are covered with a layer of white-to-gray, dense, aerial mycelia that are floccose cottony; the reverse is black with a pale brown margin.

Additional specimen examined―China, Hainan Province: the Bawangling National Nature Reserve (109°03′–109°17′ E, 18°57′–19°11′ N), on the dead branches of unidentified trees collected in tropical rain forests dominated by *Lauraceae* and *Fagaceae* trees, 19 May 2021, R.Y. Liu, paratype HMAS 352234, ex-paratype living culture SAUCC 1343.

Notes―Strains SAUCC 1342 and SAUCC 1343 have similar morphological features and identical DNA sequences and form a monophyletic group with a long branch and robust support values (MLBV = 100% and BIPP = 1.00, Figure 1). Therefore, SAUCC 1342 and SAUCC 1343 are identified as the same new species: *Acrodictys bawanglingensis.*

#### 3.2.2. *Acrodictys diaoluoshanensis* S. Wang, J.W. Xia, X.Y. Liu, and X.G. Zhang, sp. nov.

MycoBank No. 844583

Etymology―The epithet *diaoluoshanensis* pertains to the Diaoluoshan National Nature Reserve, where the holotype was collected.

Type―China, Hainan Province: the Diaoluoshan National Nature Reserve (109°41′–110°4′ E, 18°38′–18°50′ N), on the dead branches of unidentified trees collected in tropical rain forests dominated by *Lauraceae* and *Fagaceae* trees, 21 May 2021, R.Y. Liu, holotype HMAS 352235, ex-holotype living culture SAUCC 1601.

Description―Asexual morph on PDA: Conidiophores are macronematous, mononematous, erect, unbranched, straight or flexuous, thick-walled, smooth, pale brown, septate, and 34.0–65.0 × 1.8–5.6 μm. Conidiogenous cells are monoblastic, integrated, terminal, determinate, cylindrical, pale brown, and smooth. Conidia are solitary, muriform, obovoid to obpyriform, pale brown to brown, and 18.0–22.0 × 10.0–13.0 μm, usually with 3 transverse septa and 1–2 longitudinal septa; they are slightly constricted at the septa, with conspicuous pores in the septa, and truncate at the base.

Culture characteristics―Colonies on PDA are flat with an entire margin, attaining 26.0–30.0 mm in diameter after 14 days at 25 °C, with a growth rate of 1.8–2.2 mm/day; they have aerial mycelia that are white to pale brown and floccose cottony; the reverse is black. Colonies on MEA are flat with an entire margin, with white-to-gray aerial mycelia that are floccose cottony; the reverse is black with a pale brown margin.

Additional specimen examined―China, Hainan Province: the Diaoluoshan National Nature Reserve (109°41′–110°4′ E, 18°38′–18°50′ N), on the dead branches of unidentified trees collected in tropical rain forests dominated by *Lauraceae* and *Fagaceae* trees, 21 May 2021, R.Y. Liu, paratype HMAS 352236, ex-paratype living culture SAUCC 1602.

Notes―Strains SAUCC 1601 and SAUCC 1602 have similar morphological features and identical DNA sequences and gather together with robust support values (MLBV = 100% and BIPP = 1.00, Figure 1). These two strains are, therefore, identified as the same new species: *Acrodictys diaoluoshanensis*. In the phylogenetical tree, based on a combined dataset of three genetic markers, *A. diaoluoshanensis* is closely related to *A. bambusicola* (MLBV = 100%, BIPP = 1.00), but they are different in conidia (*A. diaoluoshanensis* 18.0–22.0 × 10–13 μm vs. *A. bambusicola* 17.0–36.0 × 12.0–18.0 μm).

#### 3.2.3. *Acrodictys ellisii* S. Wang, J.W. Xia, X.Y. Liu, and X.G. Zhang, sp. nov.

MycoBank No. 844584

Etymology―The epithet *ellisii* is named in honor of the great English mycologist, M. B. Ellis.

Type―China, Hainan Province: the Bawangling National Nature Reserve (109°03′–109°17′ E, 18°57′–19°11′ N), on the dead branches of unidentified trees collected in tropical rain forests dominated by *Lauraceae* and *Fagaceae* trees, 19 May 2021, R.Y. Liu, holotype HMAS 352237, ex-holotype living culture SAUCC 1471.

Description―Asexual morph on PDA: Conidiophores are macronematous, mononematous, erect, unbranched, straight or flexuous, thick-walled, smooth, pale brown, septate, and 47.0–82.0 × 2.1–5.2 μm. Conidiogenous cells are monoblastic, integrated, terminal, determinate, cylindrical, pale brown, and smooth. Conidia are solitary, muriform, obovoid to obpyriform, pale brown to brown, and 17.0–22.0 × 11.0–14.0 μm, usually with 3 transverse septa and 1–3 longitudinal septa; they are slightly constricted at the septa and truncate at the base.

Culture characteristics―Colonies on PDA are flat with an entire margin, attaining 25.0–30.0 mm in diameter after 14 days at 25 °C, with a growth rate of 1.8–2.1 mm/day; they have aerial mycelia that are white to pale brown and floccose cottony; the reverse is black. Colonies on MEA are flat with an entire margin, with white-to-gray aerial mycelia that are floccose cottony; the reverse is black with a pale brown margin.

Additional specimen examined―China, Hainan Province: the Bawangling National Nature Reserve (109°03′–109°17′ E, 18°57′–19°11′ N), on the dead branches of unidentified trees collected in tropical rain forests dominated by *Lauraceae* and *Fagaceae* trees, 19 May 2021, R.Y. Liu, paratype HMAS 352238, ex-paratype living culture SAUCC 1472.

Notes―Strains SAUCC 1471 and SAUCC 1472 are similar in morphological features and identical DNA sequences and form a clade (MLBV = 97%, Figure 1). These two strains are, therefore, identified as the same new species: *Acrodictys ellisii*. Phylogenetically, *A. ellisii* is closely related to *A. bawanglingensis* (MLBV = 97%, BIPP = 1.00), having only 4 and 18 bp of dissimilarity in LSU and SSU, respectively. Morphologically, *Acrodictys bawanglingensis* also differs from *A. ellisii* in conidial size (60.0–120.0 × 4.5–6.5 μm vs. 47.0–82.0 × 2.1–5.2 μm).

#### 3.2.4. *Acrodictys pigmentosa* S. Wang, J.W. Xia, X.Y. Liu, and X.G. Zhang, sp. nov.

MycoBank No. 844585

Etymology―The epithet *pigmentosa* originates from its pigmented colony on MEA.

Type―China, Hainan Province: the Bawangling National Nature Reserve (109°03′–109°17′ E, 18°57′–19°11′ N), on the dead branches of unidentified trees collected in tropical rain forests dominated by *Lauraceae* and *Fagaceae* trees, 19 May 2021, R.Y. Liu, holotype HMAS 352239, ex-holotype living culture SAUCC 1591.

Description―Asexual morph on PDA: Conidiophores are macronematous, mononematous, erect, unbranched, straight or flexuous, thick-walled, smooth, pale brown, septate, and 4.5–75.0 × 1.5–3.0 μm. Conidiogenous cells are monoblastic, integrated, terminal, determinate, cylindrical, pale brown, and smooth. Conidia are solitary, muriform, obovoid to obpyriform, pale brown to brown, 12.0–24.0 × 7.0–12.0 μm, usually with 1–4 transverse septa and 1–3 longitudinal septa, slightly constricted at the septa, and truncate at the base.

Culture characteristics―Colonies on PDA are flat with an entire margin, attaining 27.0–32.0 mm in diameter after 14 days at 25 °C, with a growth rate of 1.5–2.5 mm/day; they have white-to-pale-brown aerial mycelia that are floccose cottony; the reverse is black. Colonies on MEA are flat with an entire margin, with white-to-gray aerial mycelia that are floccose cottony; the reverse is black with a pale brown margin.

Additional specimen examined―China, Hainan Province: the Bawangling National Nature Reserve (109°03′–109°17′ E, 18°57′–19°11′ N), on the dead branches of unidentified trees collected in tropical rain forests dominated by *Lauraceae* and *Fagaceae* trees, 19 May 2021, R.Y. Liu, paratype HMAS 352240, ex-paratype living culture SAUCC 1592.

Notes―Strains SAUCC 1591 and SAUCC 1592 are similar in morphological features and identical DNA sequences and form a monophyletic group with robust support values (MLBV = 100%, BIPP = 1.00, Figure 1). These two strains are, therefore, identified as the same new species: *Acrodictys pigmentosa*. Phylogenetic analyses on a combined dataset of three genetic markers showed that *A. pigmentosa* is basal to the clade of *A. ellisii*, *A. bawanglingensis*, *A. bambusicola*, and *A. diaoluoshanensis* (MLBV = 96%, BIPP = 1.00), but they are different in conidia (12.0–24.0 × 7.0–12.0 μm vs. 17.0–22.0 × 11.0–14.0 μm vs. 18.0–26.0 × 10.0–16.0 μm vs. 17.0–36.0 × 12.0–18.0μm vs. 18.0–22.0 × 10.0–13.0 μm).

### 3.3. Key to the Species of Acrodictys

Together with the four new species proposed in this study, we currently accepted a worldwide total of 27 species in the genus *Acrodictys*. In order to facilitate identification in the future, a key to the species of *Acrodictys* is provided herein, updating the key compiled 11 years ago [6]. Since then, as many as eight new species have been added to this genus. The characteristics adopted in the updated key include perithecia, septa, asci, ascospores, conidiogenous cells, conidia, and chlamydospores.
1. Sexual morph known------------------------------------------------------------------------------------21’ Sexual morph unknown---------------------------------------------------------------------------------62. Maximum number of septa of ascospores > 3------------------------------------------------------32’ Maximum number of septa of ascospores ≤ 3------------------------------------------------------43. Conidia obovate---------------------------------------------------------------------------*A. satwalekeri*3’ Conidia pyriform---------------------------------------------------------------------------*A. elaeidicola*4. Conidia ellipsoid---------------------------------------------------------------------------------*A. nigra*4’ Conidia muriform-----------------------------------------------------------------------------------------55. Conidia size 23.0–34.0 × 18.0–22.0 μm----------------------------------------*A. peruamazonensis*5’ Conidia size 15.0–22.0 × 7.0–13.0 μm------------------------------------------------*A. hainanensis*6. Conidia with transverse septa only-------------------------------------------------------------------76’ Conidia with transverse and longitudinal septa-------------------------------------------------147. Conidia clavate--------------------------------------------------------------------------------------------87’ Conidia rounded-----------------------------------------------------------------------------------------108. Conidiogenous cells in groups--------------------------------------------------------------*A. similis*8’ Conidiogenous cells singly-----------------------------------------------------------------------------99. Conidiogenous cells size 70.0–100.0 × 4.0–6.0 μm------------------------------------*A. aquatica*9’ Conidiogenous cells size 98.0–142.0 × 4.0–6.0 μm---------------------------------*A. fluminicola*10. Conidiogenous cells branched---------------------------------------------------------*A. caribensis*10’ Conidiogenous cells unbranched------------------------------------------------------------------1111. Conidiogenous cells septate-------------------------------------------------------------------------1211’ Conidiogenous cells aseptate-----------------------------------------------------------------------1312. Conidia spheroid---------------------------------------------------------------------------*A. brooksiae*12’ Conidia ellipsoidal--------------------------------------------------------------------------*A. sacchari*13. Conidia 2–3 transverse septa--------------------------------------------------------------*A. elliptica*13’ Conidia 4–9 transverse septa----------------------------------------------------------------*A. liputii*14. Conidiophores in groups-------------------------------------------------------------------*A. furcata*14’ Conidiophores singly---------------------------------------------------------------------------------1515. Conidia maximum length > 100 μm-------------------------------------------*-A. septosporioides*15’ Conidia maximum length < 100 μm---------------------------------------------------------------1616. Conidiogenous cells lageniform--------------------------------------------------------------------1716’ Conidiogenous cells cylindrical--------------------------------------------------------------------2217. Conidiogenous cells determinate proliferations----------------------------------*A. balladynae*17’ Conidiogenous cells percurrent proliferations-------------------------------------------------1818. Conidia subglobose or ellipsoidal-----------------------------------------------------------------1918’ Conidia clavate or pyriform-------------------------------------------------------------------------2019. Conidia size 12.0–22.0 × 8.0–16.0 μm-----------------------------------------------*-A. irregularis*19’ Conidia size 27.0–32.0 × 12.0–16.0 μm--------------------------------------------------*A. oblonga*20. Conidiogenous cells maximum length > 200 μm ------------------------------*A. porosiseptata*20’ Conidiogenous cells maximum length < 200 μm ----------------------------------------------2121. Conidia size 20.0–30.0 × 13.0–16.0 μm---------------------------------------------*A. bambusicola*21’ Conidia size 17.0–27.0 × 11.0–15.0 μm----------------------------------------------*A. atroapicula*22. Conidiogenous cells with percurrent proliferations-------------------------------------------2322’ Conidiogenous cells with determinate proliferations-----------------------------------------2523. Conidiogenous cells maximum length > 60 μm------------------------------------*A. micheliae*23’ Conidiogenous cells maximum length < 60 μm------------------------------------------------2424. Conidia size 28.0–32.0 × 8.0–12.0 μm---------------------------------------------------*A. lignicola*24’ Conidia size 16.0–20.0 × 12.0–15.0 μm-------------------------------------------------*A. papillata*25. Conidiogenous cells maximum length > 80 μm------------------------------------------------2625’ Conidiogenous cells maximum length < 80 μm------------------------------------------------2726. Conidia size 18.0–26.0 × 10.0–16.0 μm, exceeding 23 and 13 um in length and width, respectively----------------------------------------------------------------------------***A. bawanglingensis***26’ Conidia size 18.0–22.0 × 10.0–13.0 μm----------------------------------------------------***A. ellisii***27. Conidia size 18.0–22.0 × 10.0–13.0 μm------------------------------------***A. diaoluoshanensis***27’ Conidia size 12.0–24.0 × 7.0–12.0 μm----------------------------------------------***A. pigmentosa***

## 4. Discussion

Traditionally, *Acrodictys* sensu lato species have been characterized and identified based on conidial schizolytic/rhexolytic secession, conidiophores, conidiogenous cells, and conidia [2,3,4,5]. *Acrodictys* as a single genus in *Acrodictyaceae* was introduced by Xia et al. [7] based on ITS, LSU, SSU, and tub2 sequence data. In previous studies, *Acrodictys* species have been characterized and identified based on dictyoseptate pigmented conidia seceding schizolytically from monoblastic integrated terminal determinate or lageniform to doliiform percurrently extending conidiogenous cells [1,6]. Xia et al. [7] described eight *Acrodictys* species, including a new species based on morphology and ITS, LSU, SSU, and tub2 sequence data. This makes it possible for us to study *Acrodictys* species through molecular systematics. Subsequently, Luo et al. [10] introduced a new species and a new combination for this genus. To date, 10 *Acrodictys* species have molecular data, viz. *A. aquatica*, *A. bambusicola*, *A. elaeidicola*, *A. fluminicola*, *A. globulosa*, *A. hainanensis*, *A. liputii*, *A. malabarica*, *A. peruamazonensis*, and *A. porosiseptata* [7,10].

Tropical forest ecosystems offer suitable habitats for microfungi, among which the anamorphic species are the most abundant and diverse. Many anamorphic species have been recorded in rainforests, forest parks, and national nature reserves of Hainan Province, China [23,24,25,26]. As decomposers in the ecosystem, saprobic hyphomycetes can decompose a large amount of litter in tropical rain forests and convert them into inorganic substances, returning them to the soil. During our studies of saprobic hyphomycetes from the national nature reserves of Hainan Province, particularly the regions of the Bawangling National Nature Reserve and Diaoluoshan National Nature Reserve, several collections were made on dead branches. Studies of their morphological characteristics and phylogenetic data revealed four new species. These four species share a feature in conidiogenous cells, namely determinate proliferations. The conidial characteristics provide useful information for species delimitation (Table 3). In this study, we also uploaded rpb2 and tub2 sequence data (Table 2), although few were deposited in GenBank by other experts. All the four new species reported herein are different from one another in these two loci.

## 5. Conclusions

Hainan Province has a typical tropical rainforest climate, which is very suitable for the growth and reproduction of saprotrophic fungi. In this study, we chose the Bawangling National Nature Reserve and Diaoluoshan National Nature Reserve as representative sites for sample collection. Through morphological observations and molecular date analyses, we identified four new *Acrodictys* species, namely *A. bawanglingensis*, *A. diaoluoshanensis*, *A. ellisii*, and *A. pigmentosa*. The morphological descriptions and molecular data of *Acrodictys* in this study not only enrich the world’s fungal resources and diversity but also contribute materials for studies of the effects of saprobic hyphomycetes on carbon cycling in ecosystems.

## Figures and Tables

**Figure 1 jof-08-00853-f001:**
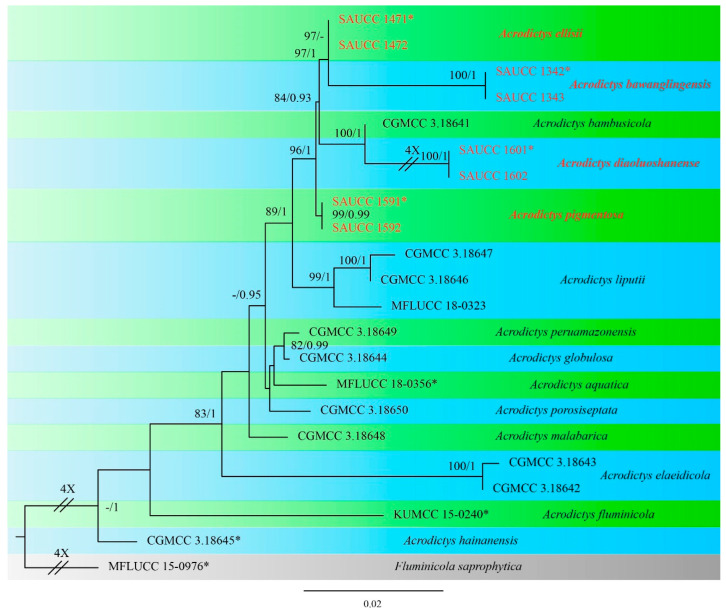
Phylogram generated from RAxML analysis based on combined ITS, LSU, and SSU sequence data, demonstrating phylogenetic relationships within *Acrodictys* with the *Fluminicola saprophytica* MFLUCC 15-0976 as outgroup. The maximum likelihood bootstrap value (MLBV ≥ 70%) and Bayesian inference posterior probability (BIPP ≥ 0.90) are shown at relative nodes at the first and second positions, respectively. Strains marked with “*” are ex-types. Strains from this study are shown in red. Three branches were shortened to fit the page size and are indicated by a double slash (//) with a fold number showing how many times they are shortened.

**Figure 2 jof-08-00853-f002:**
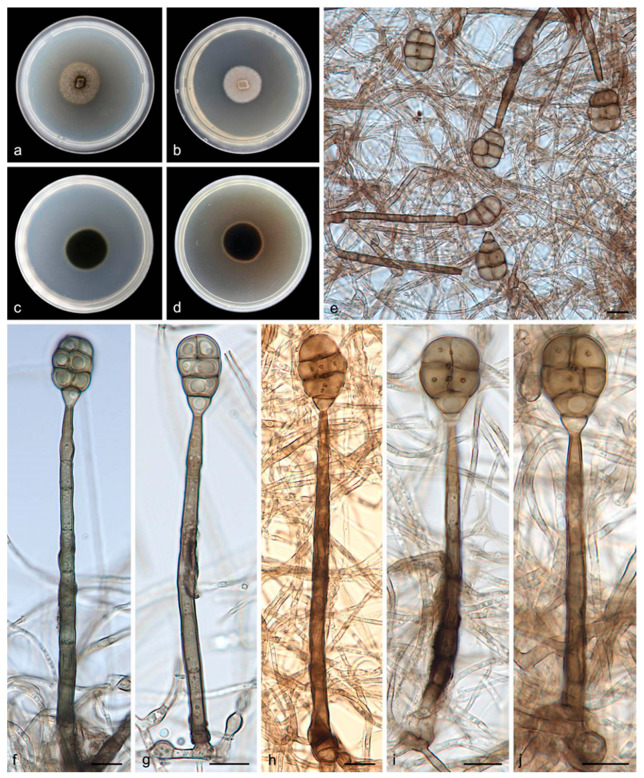
*Acrodictys bawanglingensis* (holotype HMAS 352233, ex-holotype SAUCC 1342): (**a**,**b**) surface sides of colony after incubation for 7 days on PDA (**a**) and MEA (**b**);(**c**,**d**) reverse sides of colony after incubation for 7 days on PDA (**c**) and MEA (**d**); (**e**–**j**) conidiophores, conidiogenous cells, and conidia. Petri dish diameter: 90 mm (**a**–**d**). Scale bars: 10 μm (**e**–**j**).

**Figure 3 jof-08-00853-f003:**
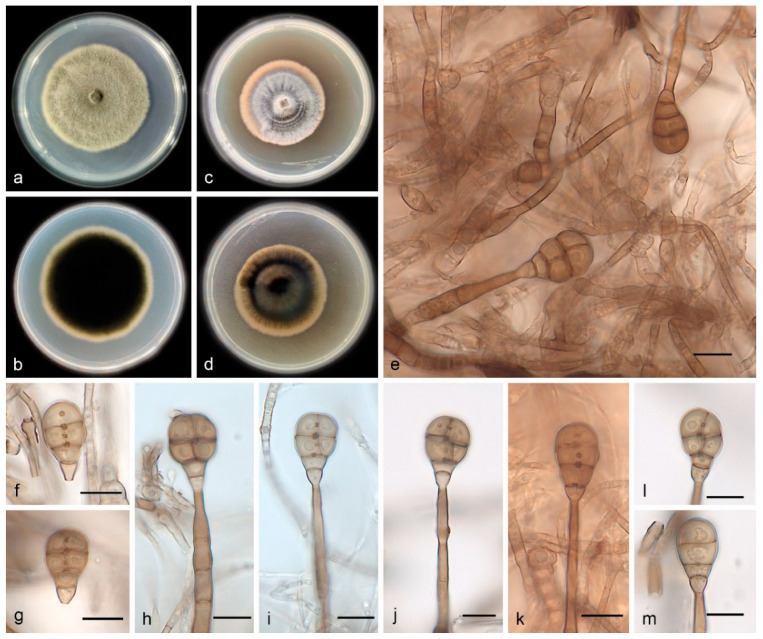
*Acrodictys**diaoluoshanensis* (holotype HMAS 352235, ex-holotype SAUCC 1601): (**a**,**b**) surface sides of colony after incubation for 30 days on PDA (**a**) and MEA (**b**); (**c**,**d**) reverse sides of colony after incubation for 30 days on PDA (**c**) and MEA (**d**); (**e**–**m**) conidiophores, conidiogenous cells, and conidia. Petri dish diameter: 90 mm (**a**–**d**). Scale bars: 10 μm (**e**–**m**).

**Figure 4 jof-08-00853-f004:**
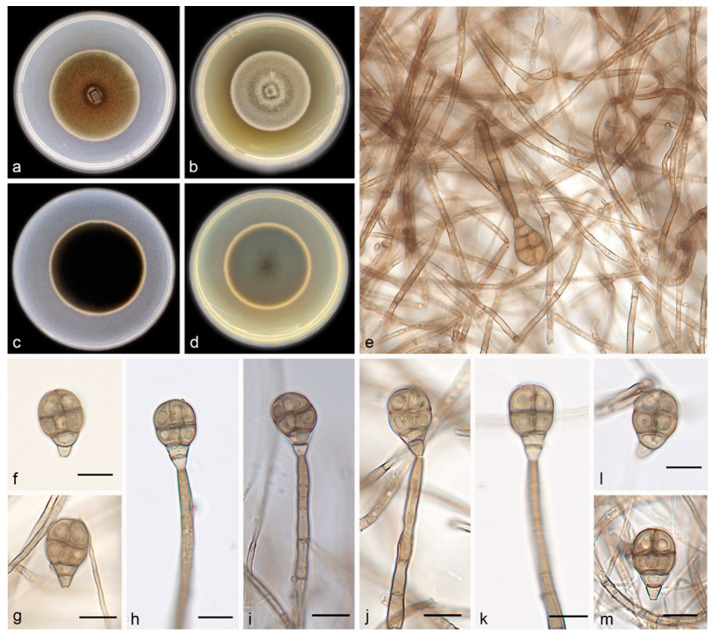
*Acrodictys ellisii* (holotype HMAS 352237, ex-holotype SAUCC 1471): (**a**,**b**) surface sides of colony after incubation for 30 days on PDA (**a**) and MEA (**b**); (**c**,**d**) reverse sides of colony after incubation for 30 days on PDA (**c**) and MEA (**d**); (**e**–**m**) conidiophores, conidiogenous cells. and conidia. Petri dish diameter: 90 mm (**a**–**d**). Scale bars: 10 μm (**e**–**m**).

**Figure 5 jof-08-00853-f005:**
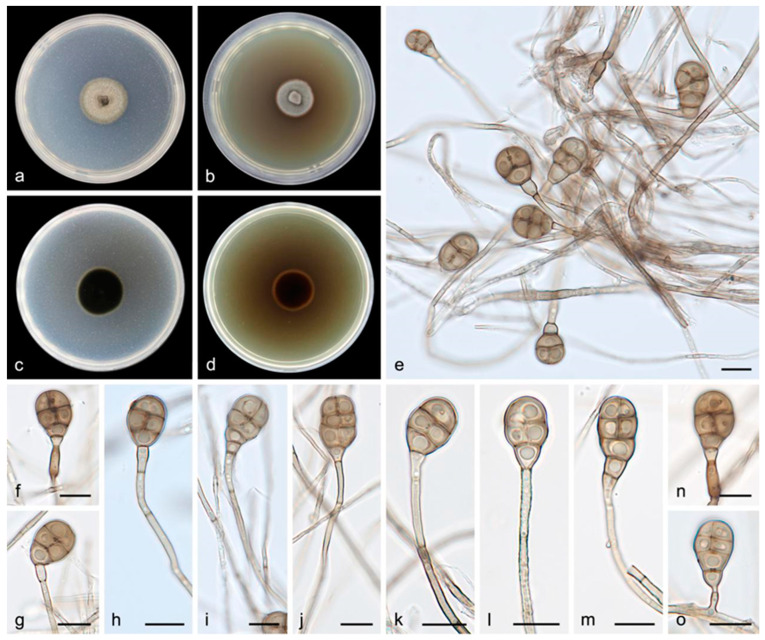
*Acrodictys pigmentosa* (holotype HMAS 352239, ex-holotype SAUCC 1591): (**a**,**b**) surface sides of colony after incubation for 7 days on PDA (**a**) and MEA (**b**); (**c**,**d**) reverse sides of colony after incubation for 7 days on PDA (**c**) and MEA (**d**); (**e**–**o**) conidiophores, conidiogenous cells, and conidia. Petri dish diameter: 90 mm (**a**–**d**). Scale bars: 10 μm (**e**–**o**).

**Table 1 jof-08-00853-t001:** Gene regions, respective primer pairs, and PCR programs used in the study.

Locus	Primer	Sequence (5′–3′)	PCR Program	References
ITS	ITS5 ITS4	GGA AGT AAA AGT CGT AAC AAG G TCC TCC GCT TAT TGA TAT GC	(94 °C for 30 s, 55 °C for 30 s, 72 °C for 45 s) × 29 cycles	[18]
LSU	LR0R LR5	GTA CCC GCT GAA CTT AAG C TCC TGA GGG AAA CTT CG	(94 °C for 30 s, 48 °C for 50 s, 72 °C for 90 s) × 35 cycles	[19,20]
SSU	NS1 NS4	GTA GTC ATA TGC TTG TCT C CTT CCG TCA ATT CCT TTA AG	(94 °C for 30 s, 50 °C for 45 s, 72 °C for 90 s) × 35 cycles	[18]
tub2	Bt-2a Bt-2b	GGT AAC CAA ATC GGT GCT GCT TTC ACC CTC AGT GTA GTG ACC CTT GGC	(95 °C for 30 s, 56 °C for 30 s, 72 °C for 60 s) × 35 cycles	[21]
rpb2	fRPB2-5F fRPB2-7R	GGG GWG AYC AGA AGA AGG C CCC ATR GCT TGY TTR CCC AT	(95 °C for 50 s, 56 °C for 50 s, 72 °C for 120 s) × 37 cycles	[22]

**Table 2 jof-08-00853-t002:** Strains and GenBank accession numbers used in this study.

Species	Strain	GenBank Accession Number				
		ITS	LSU	SSU	rpb2	tub2
*Acrodictys aquatica*	MFLUCC 18-0356 *	MG835711	MG835712	–	–	–
*A. bambusicola*	CGMCC 3.18641	KU999973	KX033564	KX033535	–	KX036219
** *A. bawanglingensis* **	**SAUCC 1342 ***	**ON606324**	**ON614219**	**ON620164**	**ON859853**	**ON859845**
	**SAUCC 1343**	**ON606325**	**ON614220**	**ON620165**	**ON859852**	**ON859844**
** *A. diaoluoshanensis* **	**SAUCC 1601 ***	**ON645265**	**ON644407**	**ON645269**	**ON859847**	**ON859839**
	**SAUCC 1602**	**ON645266**	**ON644408**	**ON645270**	**ON859846**	**ON859838**
** *A. ellisii* **	**SAUCC 1471 ***	**ON645254**	**ON644405**	**ON645267**	**ON859851**	**ON859843**
	**SAUCC 1472**	**ON645255**	**ON644406**	**ON645268**	**ON859850**	**ON859842**
*A. elaeidicola*	CGMCC 3.18642	KU999977	KX033568	KX033539	–	–
	CGMCC 3.18643	KU999978	KX033569	KX033540	–	–
*A. fluminicola*	KUMCC 15-0240 *	MK828642	MK849786	–	–	–
*A. globulosa*	CGMCC 3.18644	KU999970	KX033562	KX033532	–	–
*A. hainanensis*	CGMCC 3.18645 *	KU999974	KX033565	KX033536	–	–
*A. liputii*	CGMCC 3.18646	KU999966	KX033558	KX033528	–	–
	CGMCC 3.18647	KU999979	KX033570	KX033541	–	–
	MFLUCC 18-0323	MZ412512	MZ412524	MZ413269	–	–
*A. malabarica*	CGMCC 3.18648	KU999968	KX033560	KX033530	–	–
*A. peruamazonensis*	CGMCC 3.18649	KU999969	KX033561	KX033531	–	–
** *A. pigmentosum* **	**SAUCC 1591 ***	**ON606326**	**ON614221**	**ON620166**	**ON859849**	**ON859841**
	**SAUCC 1592**	**ON606327**	**ON614222**	**ON620167**	**ON859850**	**ON859842**
*A. porosiseptata*	CGMCC 3.18650	KU999967	KX033559	KX033529	–	KX036220
*Fluminicola saprophytica*	MFLUCC 15-0976 *	MF374358	MF374367	MF374375	MF370954	–

Notes: New species established in this study are in bold. Ex-type strains are marked with “*”.

**Table 3 jof-08-00853-t003:** Micromorphological comparison of phylogenetically related *Acrodictys* species.

Species	Shape	Conidial Size (μm)	Septa
*Acrodictys aquatica*	Clavate	20.0–27.0 × 10.0–17.0	3–4 transverse septa
*A. bambusicola*	Broadly clavate or pyriform	17.0–36.0 × 12.0–18.0	2–5 transverse and 1 or more longitudinal septa
** *A. bawanglingensis* **	**Obovoid to obpyriform**	**18.0–26.0 × 10.0–16.0**	**3 transverse and 1–3** **longitudinal septa**
** *A. diaoluoshanensis* **	**Obovoid to obpyriform**	**18.0–22.0 × 10.0–13.0**	**3 transverse and 1–2** **longitudinal septa**
** *A. ellisii* **	**Obovoid to obpyriform**	**17.0–22.0 × 11.0–14.0**	**3 transverse and 1–3** **longitudinal septa**
*A. elaeidicola*	Turbinate, pyriform or clavate	17.0–26.0 × 11.0–19.0	3 transverse and 1–3 longitudinal septa
*A. fluminicola*	Broadly clavate, obovoid to pyriform	24.0–30.0 × 13.0–17.0	2–3 transverse and a few longitudinal septa
*A. globulosa*	Subglobose	22.0–27.0 × 17.0–23.0	2 transverse and several longitudinal and oblique septa
*A. hainanensis*	Oblong to obovoid	15.0–22.0 × 7.0–13.0	3–5 transverse and several longitudinal or oblique septa
*A. liputii*	Subglobose	18.5–22.5 × 13.5–17.5	2–3 parallel transverse and 2 perpendicular longitudinal septa
*A. malabarica*	Gangliar, somewhat top-shaped	16.0–21.0 × 14.0–17.0	3 transverse and 2–4 longitudinal septa
*A. peruamazonensis*	Ellipsoidal	28.0–36.0 × 17.0–21.0	3–4 transverse and some longitudinal or oblique septa
** *A. pigmentosa* **	**Obovoid to obpyriform**	**12.0–24.0 × 7.0–12.0**	**1–4 transverse septa and 1–3 longitudinal septa**
*A. porosiseptata*	Broadly clavate to pyriform	25.0–30.0 × 13.5–16.5	4–5 transverse and 3 perpendicular longitudinal septa

Notes: New species established in this study are in bold.

## Data Availability

The sequences from the present study were submitted to the NCBI database (https://www.ncbi.nlm.nih.gov/, accessed on 10 June 2022), and the accession numbers are listed in Table 2.

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
