# Peer review of "Morphological and Phylogenetic Analyses Reveal Four New Species of Acrodictys (Acrodictyaceae) in China"

_jof, 2022, doi:10.3390/jof8080853_

Round 1
Reviewer 1 Report
The manuscript is relevant, because proposed the nov species of Acrodictys. If its possible increase the number of isolates for specie improve the quality of it.
The manuscript describes a phylogeny and morphological analysis of a series of strains isolated from Hainan China. The title suggests that the sampling area is in China, in the abstract to mark out a sampling area to Hainan. Although the research it is relevant, the objective is not clear as to whether the revision is necessary because their identification is difficult or if in view of potential new species it was deepened.
The materials and methods used are adequate. It would only be recommended to have a higher number of isolates for its development. It is not clear in results when it says that 8 strains were obtained and then L 149 refers to 22 strains.
The illustrations are clear and the key is consistent with the observed parameters. The conclusion could be improved by highlighting what is the significant contribution regarding the classification of Acrodyctis beyond the new species.
The R consider that the research is a report as communication.
Author Response
The manuscript is relevant, because proposed the nov species of Acrodictys. If its possible increase the number of isolates for specie improve the quality of it.
Response: Thanks for your suggestion. However, due to the pandemic COVID-19, we could not make more field trips and obtain more isolates. We believe we can isolate more in the future investigations.

Reviewer 2 Report
Dear Authors,
Congratulations for the discovery of four new species to science. Your methods and results are nicely presented and some minor corrections are needed. Please go through the attached document and take into consideration my comments (highlighted text with pop-up windows). Some clarification is needed at several places in the text.
Concerning citing collections (all types) I would like you to improve data with more details on habitat and localities. Nature reserves are quite big areas and more topographic information is required.
Figure containing phylogram is of very low resolution and therefore not suitable to publication. You should replace it with the one of higher quality.
Key to the species should be designed following the journal guidelines.
Sincerely

Author Response
Congratulations for the discovery of four new species to science. Your methods and results are nicely presented and some minor corrections are needed. Please go through the attached document and take into consideration my comments (highlighted text with pop-up windows). Some clarification is needed at several places in the text.
Concerning citing collections (all types) I would like you to improve data with more details on habitat and localities. Nature reserves are quite big areas and more topographic information is required.
Figure containing phylogram is of very low resolution and therefore not suitable to publication. You should replace it with the one of higher quality.
Key to the species should be designed following the journal guidelines.
Response: According to for your suggestion, we have revised the manuscript and detailed as follows.
Line 23. Please reorder the keywords alphabetically
Response: Thank you. We have revised as “morphology; new taxa; phylogeny; Sordariomycetes; taxonomy”
Line 31. “obovate” > “obovata”
Response: Thank you. We have corrected this epithet.
Line 46. “twelve” > “12”
Response: Thanks for your suggestion. We have rewritten as “12”.
Line 54. “from which pure cultures were isolated” > “from which fungi were isolated in pure cultures”
Response: Thank you. We revised as “the fungi in pure culture were isolated from these dead branches”.
Line 54. “Theses cultures” > “Fungi from these axenic cultures”
Response: Thank you. We revised as your suggestion.
Line 61. “Hannan” > “Hainan”
Response: Thank you. We have corrected this word.
Line 62. “petri” > “Petri”
Response: Thank you. We have corrected this mistake.
Line 78. “with” > “using”
Response: Thank you. We followed your suggestion.
Line 79. “version” > “ver.”
Response: Thank you. We accepted this suggestion.
Line 63. “from the” > “among”
Response: Thank you. We accepted this suggestion.
Line 82. Unclear! What does it mean "deposited in Microbiology"? Department or Laboratory or similar?
Response: Thanks for your suggestion. We have corrected this as “deposited in the Herbarium Mycologicum Academiae Sinicae, Institute of Microbiology”.
Line 90. “al” > “al.”
Response: Thank you. We have corrected this mistake.
Line 129. All mentioned "type" categories are basically TYPE. So you can say only "ex-type strains are marked with..." If you want to distinguish between holotype and epitype you should use different marks, not only one ("*"). e. g. small ET in superscript for epitype and HT in superscript for holotype
Response: Thanks for your suggestion. We decide to keep ex-type only.
Line 130. “Comparison” > “Micromorphological comparison”
Response: Thank you. We have corrected this.
Line 135. “The alignment” > “The alignment of...”
Response: Thanks. We have done this correction.
Line 140. In Mat&Met you mention 5.000.000 generations?
Response: In Materials and Methods, we set 5,000,000 generations for Bayesian Inference. This is a preset generation number. Practically, the average standard deviation of split frequencies is lower than 0.01 at the 65,000th generation, and consequently the run stopped automatically.
Line 151. Different font?
Response: Thanks for your suggestion. We have modified it.
Line 162. Please add the information on Petri dish size so that reader can calculate the size of the colony. Or simply add the scale bar also on a-d.
Response: Thanks for your suggestion. We have added Petri dish size annotations to the picture annotations and colony size descriptions in “Culture characteristics”.
Line 169. Could you please define locality more specific? Large natural reserves are quite wide area. If possible, add a coordinate taken with a GPS device or from the map. If you do not have a coordinate, try to define a locality using some precise maps which has more toponyms. Apply this comment to all collections mentioned in this article.
Response: Thanks for your suggestion. We have added GPS coordinates to every location mentioned in the article.
Line 170. Since you do not know the substrate (which species of a plant), please describe a habitat: if it is a forest mention which trees are dominant (main edificators of that forest) in that forest. Apply this comment also to all other collections.
Response: Thanks for your suggestion. Details have been added in the article.
Line 190. DNA sequences are also "similar", not the same? Please add the percentage of similarity (if not 100%) between holotype and paratype.
Response: Thanks for your suggestion. The DNA sequences of the two strains are the same and have been modified in the article.
Line 199,200. “(e‒o)” > “(e‒m)”
Response: Thank you. We have corrected.
Line 220. “exparatype” > “ex-paratype”
Response: Thank you. We have corrected this.
Line 309. Please distinguish two choices - consult the journal guidelines
Response: Thank you. The journal guidelines have been reviewed and relevant content has been revised.
Line 314. “only transverse” > “only with transverse”
Response: Thank you. We have done this correction.
Line 346,347. “with”
Response: Thank you. We have done this correction.
Line 354. Conidia exceeding 23 μm in length, reaching 26 μm; and exceeding 13 μm in width, reaching 16 μm.
Response: Thank you for your suggestion. However, for a better comparison and a more complete presentation of the information, we think it is better to list the entire size range. How do you think about our thought?
Line 375. Please use the modern term "saprobic" or "saprotrophic"
Response: Thanks for your suggestion. We have replaced the “saprophytic” with “saprobic”.
Line 386. Why only this region and not worldwide?
Response: Thanks. We have modified the expression per your suggestion.
Reviewer 3 Report
Add significance of these fungi in the natural habitat, and why they should be studied. Relevant Information to the above point can be added in both intro and discussion section of the manuscript.
Please also provide a brief conclusion of the study at the end of the discussion.
Author Response
Add significance of these fungi in the natural habitat, and why they should be studied. Relevant Information to the above point can be added in both intro and discussion section of the manuscript.
Response: Thanks. We have added importance for this group of fungi in both introduction and discussion: “They are thought to be wood decaying fungi, promoting the carbon cycle in the ecosystem by converting cellulose, hemicellulose and lignin into inorganic substances” and “As decomposers in the ecosystem, saprobic hyphomycetes can decompose a large number of litters in tropical rain forests and convert them into inorganic substances to return to the soil”.
Please also provide a brief conclusion of the study at the end of the discussion.
Response: Thanks for your suggestion. We have added a section “Conclusion” after the “Discussion” section. It reads as “Hainan Province has a typical tropical rain forest climate, which is very suitable for the growth and reproduction of saprotrophic fungi. This study chooses Bawangling Na-tional Nature Reserve and Diaoluoshan National Nature Reserve as representative sites for sample collection. Through morphological observations and molecular date analyses, we identify four new Acrodictys species, namely A. bawanglingensis, A. diaoluoshanensis, A. ellisii and A. pigmentosa. The morphological descriptions and molecular data of Acrodictys in this study not only enrich the world's fungal resources and diversity, but also contribute materials for studies of the effects of saprobic hyphomycetes on carbon cycling in ecosystems”.
